# In Silico, In Vitro, and In Vivo Investigations on Adapalene as Repurposed Third Generation Retinoid against Multiple Myeloma and Leukemia

**DOI:** 10.3390/cancers15164136

**Published:** 2023-08-16

**Authors:** Joelle C. Boulos, Manik Chatterjee, Letian Shan, Thomas Efferth

**Affiliations:** 1Department of Pharmaceutical Biology, Institute of Pharmaceutical and Biomedical Sciences, Johannes Gutenberg University, Staudinger Weg 5, 55128 Mainz, Germany; joboulos@uni-mainz.de; 2Translational Oncology, Comprehensive Cancer Center Mainfranken, University Hospital Würzburg, 97080 Würzburg, Germany; chatterjee_m@ukw.de; 3The First Affiliated Hospital, Zhejiang Chinese Medical University, Hangzhou 310053, China; letian.shan@foxmail.com

**Keywords:** drug repurposing, hematological malignancies, microtubules, programmed cell death, targeted chemotherapy, third-generation retinoid, xenograft tumor zebrafish model

## Abstract

**Simple Summary:**

Hematological neoplasms afflict millions of children and adults yearly and are often uncurable due to refractory illness and recurrence. Therefore, new treatment approaches are required. The proto-oncogene c-MYC has been linked to carcinogenesis, particularly in hematological malignancies. Consequently, to develop new and effective therapies for blood cancers, it is essential to target c-MYC, specifically that c-MYC inhibitors have not yet received clinical approval. In this research, the verification of adapalene as a c-MYC inhibitor may be a glimmer of hope, especially that it is an already Food and Drug Administration (FDA)-approved drug, and its toxicity profiles, pharmacokinetics, and pharmacodynamics are well established. Our study presents a rationale that the discovery of adapalene as a c-MYC inhibitor may significantly lower the drug development costs of new anticancer medications. It also provides further insights in the future of adapalene-based designs that could result in more effective and targeted innovative therapies for multiple myeloma.

**Abstract:**

The majority of hematopoietic cancers in adults are incurable and exhibit unpredictable remitting-relapsing patterns in response to various therapies. The proto-oncogene c-MYC has been associated with tumorigenesis, especially in hematological neoplasms. Therefore, targeting c-MYC is crucial to find effective, novel treatments for blood malignancies. To date, there are no clinically approved c-MYC inhibitors. In this study, we virtually screened 1578 Food and Drug Administration (FDA)-approved drugs from the ZINC15 database against c-MYC. The top 117 compounds from PyRx-based screening with the best binding affinities to c-MYC were subjected to molecular docking studies with AutoDock 4.2.6. Retinoids consist of synthetic and natural vitamin A derivatives. All-trans-retinoic acid (ATRA) were highly effective in hematological malignancies. In this study, adapalene, a third-generation retinoid usually used to treat acne vulgaris, was selected as a potent c-MYC inhibitor as it robustly bound to c-MYC with a lowest binding energy (LBE) of −7.27 kcal/mol, a predicted inhibition constant (pKi) of 4.69 µM, and a dissociation constant (K_d_ value) of 3.05 µM. Thus, we examined its impact on multiple myeloma (MM) cells in vitro and evaluated its efficiency in vivo using a xenograft tumor zebrafish model. We demonstrated that adapalene exerted substantial cytotoxicity against a panel of nine MM and two leukemic cell lines, with AMO1 cells being the most susceptible one (IC_50_ = 1.76 ± 0.39 µM) and, hence, the focus of this work. Adapalene (0.5 × IC_50_, 1 × IC_50_, 2 × IC_50_) decreased c-MYC expression and transcriptional activity in AMO1 cells in a dose-dependent manner. An examination of the cell cycle revealed that adapalene halted the cells in the G_2_/M phase and increased the portion of cells in the sub-G_0_/G_1_ phase after 48 and 72 h, indicating that cells failed to initiate mitosis, and consequently, cell death was triggered. Adapalene also increased the number of p-H3(Ser10) positive AMO1 cells, which is a further proof of its ability to prevent mitotic exit. Confocal imaging demonstrated that adapalene destroyed the tubulin network of U2OS cells stably transfected with a cDNA coding for α-tubulin-GFP, refraining the migration of malignant cells. Furthermore, adapalene induced DNA damage in AMO1 cells. It also induced apoptosis and autophagy, as demonstrated by flow cytometry and western blotting. Finally, adapalene impeded tumor growth in a xenograft tumor zebrafish model. In summary, the discovery of the vitamin A derivative adapalene as a c-MYC inhibitor reveals its potential as an avant-garde treatment for MM.

## 1. Introduction

Cancers affecting lymph nodes, bone marrow, and blood are collectively known as hematological malignancies. This category comprises lymphoma (Hodgkin and non-Hodgkin types), multiple myeloma (MM), and different kinds of leukemia such as chronic myeloid (CML), acute myeloid (AML), chronic lymphocytic (CLL), and acute lymphocytic (ALL). All together, these hematopoietic neoplasms constitute about 9% of all cancers that are newly diagnosed, with lymphomas (particularly non-Hodgkin’s) being more prevalent than MM or leukemia. Blood cancers often increase with age, with the exception of Hodgkin’s lymphoma and ALL [1].

MM is a rare blood malignancy that accounts for 1% of all cancers and 10% of hematological cancers. It is the second most frequent blood cancer in industrialized countries. A hallmark of MM is the invasion of the bone marrow by monoclonal plasma cells which produce and discharge monoclonal immunoglobulin into the blood and urine [2]. The amassment of these immunoglobulins causes organ malfunction such as bone lesions, anemia, renal insufficiency, and hypercalcemia. At this point, patients begin to develop symptoms. MM generally starts as an asymptomatic precursor malignancy. Monoclonal gammopathy of undetermined significance (MGUS) and smoldering undetermined significance (MGUS) are the diseases that manifest at this stage in which genetic defects, such as hyperdiploidy and the involvement of immunoglobulin heavy chain in translocations, are initiated [3]. A translocation adjoining an oncogene next to an immunoglobulin enhancer occurs in the plasma cells of around half of individuals with MGUS [4]. These translocations might serve as a starting point for later oncogenic processes that eventually engender MM. Malignant cell gene expression profiling revealed diverse disease subgroups, likely attributable to the activation of several oncogenic pathways in various patients [5].

c-MYC deregulation may contribute to the progression of MGUS to MM [6]. Several processes, including mutation, translocation, and gene amplification, result in c-MYC overexpression. Numerous MM cell lines and, to a certain extent, primary MM cells (15–46%) harbor c-MYC chromosomal rearrangements [7]. Additionally, 40% of all c-MYC translocations in human MM cells do not affect immunoglobulin genes. Contrarily, patients overexpressing c-MYC have been delineated in a greater portion than the ones with c-MYC translocations [8]. Importantly, the experimental c-MYC knockdown was lethal to MM cells, implying that MM may be addicted for c-MYC activity to survive, and targeting c-MYC may improve the treatment outcome of MM [9].

Even with recent medical advancements such as monoclonal antibodies, histone deacetylase inhibitors, immunomodulatory drugs, proteasome inhibitors, anthracyclines, alkylating agents, corticosteroids, stem cell transplantation, radiotherapy, bisphosphonate treatment, and combination therapies [10], patients with MM frequently experience multiple relapses following one or more treatment procedures, or evolve into a refractory state, primarily as a result of drug resistance. Having in mind the impending negative effects of the available hematopoietic treatments and given that c-MYC dysregulation is a distinct feature in the genetic profile of MM, alternative approaches were forged to identify novel drugs targeting c-MYC, especially because such therapies are not yet available [11,12]. To date, numerous approaches have been developed to specifically target c-MYC in MM. Nevertheless, the pharmacokinetic characteristics and the efficacy of many of these strategies are subpar. Moreover, finding ligands that directly inhibit c-MYC is still challenging [13].

Despite improvements in technology and the advanced perception of human diseases, the development of new therapeutic approaches has proceeded far slower than anticipated. Drug repurposing, often referred to as drug reprofiling, repositioning, or re-tasking, could be an alternate method for discovering new applications for approved or experimental drugs that go beyond the limits of the typical medical prescription [14]. This strategy has significant advantages over developing a brand-new medication for a specific indication because the toxicity profiles, pharmacokinetics, and pharmacodynamics of commercially available drugs are well established. If repurposing is successful, it is possible to significantly save time and financial outlays on a drug’s preclinical and clinical testing [15].

Adapalene (Figure 1A), also known as CD271 and differin, belongs to the third generation of retinoids that were approved by the Food and Drug Administration (FDA) in 1996 to treat acne vulgaris [16]. Adapalene is a naphthoic acid derivative known to bind to retinoic acid nuclear receptors [17]. There are several documented actions of adapalene, including immunomodulatory, anti-proliferative, anti-inflammatory, comedolytic, anti-bacterial, and neuroprotective activities [18]. Regarding anticancer studies, adapalene promoted G1 cell cycle arrest and inhibited CDK2 in colorectal cancer cells. It also prompted apoptotic cell death in hepatoma cells by modulating the ratio of Bax and Bcl-2 [17,19]. Furthermore, adapalene inhibited the proliferation of melanoma cells through S-phase arrest of the cell cycle, and subsequently induced apoptosis through DNA damage [20]. Adapalene suppressed the proliferation of ES-2 ovarian cancer cells by inhibiting glutamicoxaloacetic transaminase 1 (GOT1) [21]. It also reduced the growth of prostate cancer cells by triggering DNA damage, halting the cell cycle in the S phase, and evoking apoptosis [22]. Finally, adapalene arrested triple-negative breast cancer cells in the S-phase, inhibiting their proliferation [23]. All these findings proved that repurposing adapalene for cancer treatment represents a promising approach.

The aim of the present study is to locate prospective FDA-approved drugs that target the oncogenic transcription factor c-MYC using in silico tools (virtual screening by PyRx and molecular docking by AutoDock) to examine the molecular interaction of the preeminent FDA-approved drug adapalene and c-MYC using microscale thermophoresis (MST), and to confirm that adapalene is an effective c-MYC inhibitor by a MYC reporter assay. Given that MM cells depend on c-MYC for their survival, and since there are currently no reports on the application of adapalene for the treatment of hematological malignancies, we were further interested in investigating the effect of adapalene against a panel of MM cell lines, determining its mechanism of action and its cell death modality through in vitro studies. Finally, we verified its growth inhibitory efficacy in vivo using a T-ALL zebrafish model.

## 2. Materials and Methods

### 2.1. Virtual Drug Screening and Molecular Docking Analyses

The FDA-approved drug library, consisting of 1577 compounds, was downloaded from the Zinc15 database (https://zinc15.docking.org/, accessed on 25 April 2019) to evaluate the in silico binding strength of those compounds toward c-MYC using PyRx0.9 virtual screening tool and AutoDock 4.2.6 molecular docking.

The 3D structures of the ligands were downloaded as standard data files (sdf). The crystal structure of c-MYC was downloaded from the Protein Data Bank (http://www.rcsb.org/, accessed on 25 April 2019) as a PDB file (PDB ID: 1NKP) [24]. Using AutoDockTool 1.5.6 (The Scripps Research Institute, La Jolla, CA, USA), the protein crystal structure was subsequently improved by removing water molecules and by adding missing hydrogen atoms. Following that, the PDB file was converted to Protein Data Bank Partial Charge and Atom Type (PDBQT) file [25]. Prior to PyRx running, the energy of the FDA-approved ligands was minimized, and the files were converted to PDBQT. The ligands were then sorted according to their lowest binding energy (kcal/mol), which was obtained by the PyRx virtual screening tool.

The top 170 compounds with the lowest binding energy were selected for further molecular docking analysis using AutoDock 4.2.6 (Center for Computational Structural Biology ccsb, La Jolla, CA, USA) [25]. Two known c-MYC inhibitors, 10058-F4 and 10074-G5, were used as positive controls [26,27]. The grid box was set to mask the whole protein with the center of the grid box at x = 65.333, y = 62.507, and z = 42.943 with the number of grid points (npts) of 54 in x, 108 in y, and 70 in z. The calculations were performed using a Lamarckian algorithm provided by AutoDock with 250 runs and 2,500,000 energy evaluation. The AutoDock RMSD cluster analysis generated the expected results as DLG files. Interacting with amino acids and visualizations was performed with BIOVIA Discovery Studio Visualizer (https://discover.3ds.com/, accessed on 20 May 2019).

### 2.2. Microscale Thermophoresis

MST was performed to study the in vitro binding of adapalene to recombinant human c-MYC protein (ab169901, Abcam, Cambridge, UK) [28]. Briefly, the Protein Labeling Kit Blue-NHS (L003, NanoTemper Technologies GmbH, Munich, Germany) was used to fluorescently tag the c-MYC protein following the manufacturer’s instructions. With a final c-MYC concentration of 200 nM, MST was performed at 50% LED intensity and 40% MST power. Finally, the fit curve and the dissociation constant (K_d_) of adapalene were generated with the NanoTemper Analysis Software version 2.2.4.

### 2.3. MYC Cignal Reporter Assay

The Qiagen cignal MYC reporter assay kit (CCS-012L, Germantown, MD, USA) was used to examine the impact of adapalene on c-MYC activity, as previously described [29]. Briefly stated, HEK293 human embryonic kidney cells were transfected with a c-MYC-luciferase reporter construct and grown in accordance with the instructions provided by the manufacturer. Cells were then exposed to adapalene (5 and 10 µM), DMSO as negative control, or 10058-F4 (127.5 2 µM) as positive control. After 24 h, the Dual-glo^®^ Luciferase Reporter Assay System (E2920, Promega, Madison, WI, USA) was added to each well to quantify the activity of the c-MYC promoter by measuring the luminescence of firefly and renilla luciferases using an Infinite M2000 Pro^TM^ plate reader (Tecan, Crailsheim, Germany).
$$c-Myc\;activity=\frac{firefly\;luciferase\;luminescence}{renilla\;luciferase\;luminescence}$$
$$Relative\;luciferase=100\times \left(\frac{firefly\;luciferase\;luminescence}{renilla\;luciferase\;luminescence}\right)$$
$$Normalized\;c-Myc\;activity=\frac{relative\;luciferase\;(sample)}{relative\;luciferase\;(DMSO)}$$

### 2.4. Cell Lines

Leukemia cells, including the drug-sensitive CCRF-CEM T-cell acute lymphoblastic leukemia (T-ALL) and their multidrug-resistant P-glycoprotein-overexpressing CEM/ADR5000 subline, were provided by Dr. Axel Sauerbrey (Children’s Hospital, University of Jena, Jena, Germany). HL60 and U266 were provided by Prof. Markus Munder (University Medical Center of the Johannes Gutenberg University, Mainz, Germany). MM cells, including AMO1, KMS12BM, MolP8, NCI-H929, OPM2, KMS11, L363, and JJN3, were kindly supplied by Dr. Manik Chatterjee (University of Würzburg, Würzburg, Germany). RPMI8226 cells were obtained from the American Type Cell Culture Collection (ATCC^®^ CCL-155™, ATCC, Manassas, VA, USA).

Leukemia cells and MM were maintained in RPMI 1640 (Life Technologies, Darmstadt, Germany), supplemented with 10% FBS (Life Technologies) and 1% penicillin (1000 U/mL)/streptomycin (100 μg/mL) (Life Technologies). Cells were incubated in a 5% CO_2_/37 °C incubator.

HEK293 were kindly provided by Prof. Dr. Christina Friedland (Johannes Gutenberg University, Mainz, Germany). Human bone osteosarcoma cells (U2OS) that persistently express an α-tubulin-GFP construct were kindly provided by Dr. Joachim Hehl (Light microscope center, ETH Zurich, Zurich, Switzerland). HEK293 and U2OS were maintained in DMEM (Life Technologies), supplemented with 10% FBS (Life Technologies) and 1% penicillin (1000 U/mL)/streptomycin (100 μg/mL) (Life Technologies). Cells were incubated in a 5% CO_2_/37 °C incubator.

Fresh blood samples were obtained from a healthy individual and placed in plastic Monovette EDTA tubes at the Department of Hematology, Oncology, and Pneumology (University Medical Center of the Johannes Gutenberg University, Mainz, Germany).

Histopaque^®^ (Sigma-Aldrich, Taufkirchen, Germany) was used to collect human peripheral blood mononuclear cells (PBMCs). In a few words, 3 mL of fresh blood was cautiously deposited on top of Histopaque^®^ and centrifuged at 400× *g*/4 °C for 30 min. The buffy coat made up of PBMCs was then isolated, rinsed with PBS, and centrifuged three times at 250 × *g*, for 10 min each. The resulting cell pellet was cultured in Panserin 413 growth media (PAN-Biotech, Aidenbach, Germany) and supplemented with 2% phytohemagglutinin M (PHA-M, Life Technologies).

### 2.5. Cell Viability Assessment

The sensitivity of MM cells, leukemia cells, and PBMCs to adapalene was evaluated by the resazurin reduction assay, as previously described [30]. In brief, 10^4^ cells were seeded in each well of a flat bottom 96-well plate. Cells were directly treated with 10 different concentrations of adapalene (Activate Scientific, Prien am Chiemsee, Bavaria) which are 3-fold apart from one another, ranging from 100 µM to 0.003 µM for MM and T-ALL cells. However, when 10-fold apart from one another, they ranged from 100 µM to 10^−7^ µM for PBMCs. The plates were incubated for 72 h at 5% CO_2_/37 °C and then re-incubated for another 4 h under the same conditions, with 20 μL of resazurin (0.01% *w*/*v*; Sigma-Aldrich) added to each well. An Infinite M2000 Pro plate reader (Tecan, Crailsheim, Germany) was used to measure resorufin fluorescence, produced by the reduction of resazurin by live cells, at 544–590 nm (excitation-emission wavelengths). Afterward, cell viability was plotted vs. adapalene concentration, and the IC_50_ values from three separate experiments with six repeats each were calculated with GraphPad Prism 5 software (GraphPad Software, San Diego, CA, USA).

### 2.6. Cell Cycle Analysis

The cell cycle progression of AMO1 cells incubated with different adapalene concentrations (0.5 × IC_50_, 1 × IC_50_, or 2 × IC_50_) or media for 48 and 72 h was examined by propidium iodide (PI) staining (Thermo Fisher Scientific, Dreieich, Germany). Cells were collected and maintained at −20 °C in 80% ethanol. Following the corresponding incubation time, cells were kept in PI staining solution at 4 °C for 15 min. Later on, the amount of PI staining was assessed with an Accouri C6 flow cytometer (Becton-Dickinson, Heidelberg, Germany). Total DNA content was calculated on FL2-A [31,32].

### 2.7. Confocal Microscopy of p-H3(Ser10)

AMO1 cells were incubated with adapalene (1 × IC_50_ or 2 × IC_50_) or DMSO at 5% CO_2_/37 °C for 48 h. Afterwards, cells were harvested, washed with PBS, and cytospinned to glass slides (Thermo Fisher Scientific, Dreieich, Germany) for 5 min at 1000 rpm. Cells were then fixed with 4% paraformaldehyde at room temperature for 15 min, washed twice with PBS, and permeabilized with 1% Triton X-100 for 10 min at room temperature. Following permeabilization, cells were again washed twice with PBS. Subsequently, blocking buffer (1% BSA + 10% FBS in PBS) was applied to the cells for 1 h. After blocking, the slides were coated with the primary antibody anti-phospho-histone H3 (Ser10) clone 3H10, FITC Conjugate (Merck, Darmstadt, Germany), at a concentration of 4 µg/mL. The slides had been stored in a humid environment at room temperature. After 1 h of staining, the slides were washed three times with PBS, and the cell nuclei were stained with 1 µg/mL of 4′,6-diamidino-2-phenylindole (DAPI) (Sigma-Aldrich, Darmstadt, Germany) for 5 min. Slides were then washed three times with PBS. Finally, using an AF7000 widefield fluorescent microscope (Leica Microsystems, Wetzlar, Germany), cells were examined after being coated with Fluoromount-G^®^ (SouthernBiotech, Birmingham, AL, USA). FITC was observed at 470/525 nm (excitation/emission wavelengths). DAPI was observed at 470/447 nm (excitation/emission wavelengths) [15].

### 2.8. Fluorescence Microscopy Imaging of Microtubules Structure

U2OS human osteosarcoma cells persistently expressing α-tubulin-GFP were grown in a µ-Slide 8 Well (30,000 cells/well) (ibidi, Gräfelfing, Germany). Cells were incubated overnight in a 5% CO_2_/37 °C incubator. Afterwards, cells were exposed to different adapalene concentrations (1 × IC_50_ or 2 × IC_50_) or media for 48 h. Cells were then rinsed with PBS and Hoechst 33342 Nuclear Stain (BioVision, Wiesbaden, Germany) was added for 30 min at room temperature in the dark to stain cells nuclei. Later on, excess Hoechst stain was removed with PBS, and the slides were coated with Fluoromount-G^®^. Living cells were imaged with AF7000 widefield fluorescence microscope (Leica Microsystems, Wetzlar, Germany). GFP was observed at 470/525 nm (excitation/emission wavelengths). Hoechst stain was observed at 470/447 nm (excitation/emission wavelengths). Finally, the software Fiji ImageJ version 1.8.0_322 (National Institutes of Health, Bethesda, MD, USA) was used to analyze images [33].

### 2.9. Apoptosis Assessment

Apoptosis was examined using fluorescein isothiocyanate (FITC)-conjugated annexin V/propidium iodide (PI) assay kit (eBioscience^TM^ Annexin V; Invitrogen, San Diego, CA, USA). Briefly, 10^6^ of AMO1 cells were seeded in each well of a 6-well plate. Cells were directly treated with three different concentrations of adapalene (0.5 × IC_50_, 1 × IC_50_, 2 × IC_50_) or DMSO (used as solvent control), or the positive control bortezomib (1µM) for 48 and 72 h. Subsequently, cells were collected and washed twice; the first time with cold PBS and the second time with annexin binding buffer (1×). After centrifuging the cells at 1500 rpm, 5 μL of fluorochrome-conjugated annexin V were applied to the cells, and the mixture was then incubated at room temperature for 15 min. PI staining solution (2.5 µL of PI diluted in 400 µL binding buffer (1×)) was used to stain the cells. Apoptosis was assessed using an Accuri C6 flow cytometer (Becton-Dickinson, Heidelberg, Germany). A 530/30 nm band pass filter was used to detect the annexin V-FITC signal after being excited at 488 nm. A 610/20 nm band pass filter was used to detect PI signal after being excited at 561 nm. The F- (forward) and S- (side) scatters gated living cells and the A- (area) and W- (width) scatters gated singlets. 10^4^ events/sample were documented from the F- (forward) and S- (side) scatters. Cytograhs were obtained with with FlowJo V10.6.2 software (Becton Dickinson) [34].

### 2.10. Autophagy Examination

Autophagy was assessed using the autophagy detection kit (ab139484, Abcam, Cambridge, UK). Briefly, 10^6^ of AMO1 cells were seeded in each well of a 6-well plate. Cells were directly treated with 3 different concentrations of adapalene (0.5 × IC_50_, 1 × IC_50_, 2 × IC_50_) or DMSO (used as solvent control) for 48 and 72 h. The positive control rapamycin (500 nM) was added 18 h prior to detection. Subsequently, cells were collected, washed with PBS, and suspended in 250 µL of PBS supplemented with 5% FBS. Following that, 250 µL of the diluted green stain solution (1 µL of green detection reagent was diluted in 999 µL of PBS supplemented with 5% FBS) was added to each sample for 30 min at 37 °C in dark. Afterwards, the cells were collected by centrifugation, washed with 1× assay buffer, and resuspended with 500 µL of 1× assay buffer. Autophagy was assessed using an Accouri C6 flow cytometer (Becton-Dickinson, Heidelberg, Germany). A 530/30 nm band pass filter was used to detect the green detection reagent after being excited at 488 nm. The F- (forward) and S- (side) scatters gated living cells and the A- (area) and W- (width) scatters gated singlets. 10^4^ events/sample were documented from the F- (forward) and S- (side) scatters. Cytographs were obtained with FlowJo V10.6.2 software (Becton-Dickinson, Heidelberg, Germany) [35].

### 2.11. Western Blot

Aliquots of 10^6^ of AMO1 cells were seeded in each well of a 6-well plate and subjected to various concentrations of adapalene (0.5 × IC_50_, 1 × IC_50_, or 2 × IC_50_) or with the negative control DMSO for 48 h at 5% CO_2_/37 °C. Cells were then collected and washed twice with PBS. Total protein extraction was accomplished by suspending the cells in M-PER^®^ Mammalian Protein Extraction Reagent supplemented with 1% Halt™ Protease Inhibitor Cocktail (Thermo Scientific, Frankfurt, Germany) and incubated in a 500-rpm shaker at 4 °C for 30 min. Afterwards, the proteins in the supernatant were collected by centrifuging the cells at 1500 rpm at 4 °C for 15 min. A NanoDrop 1000 spectrophotometer (Thermo Scientific, Frankfurt, Germany) was used to determine protein concentration. Following that, 30 µg of proteins were loaded to each loading well of an SDS-PAGE gel (10%). Proteins were transferred onto a polyvinylidene difluoride (PVDF) membrane after being separated by migration in the separating gel. The membrane was then soaked in blocking buffer (5% BSA in TBST) at room temperature for 1 h. Subsequently, membranes were incubated at 4 °C overnight with primary antibodies (1:1000) against c-MYC (#9402), Phospho-Histone H2A.X (Ser139) (20E3) (#9718), GAPDH (D16H11) XP^®^ (#5174), and β-Actin (13E5) (#4970), which were obtained from Cell Signaling Technology (Frankfurt a. M., Germany), and P62 (18420-1-AP), which was obtained from Proteintech (Planegg-Martinsried, Germany). After being washed three times with TBST for 10 min each, membranes were incubated with the proper secondary antibody conjugated with horseradish peroxidase (1:1000) at room temperature for 1 h. Lastly, membranes were photographed using an Alpha Innotech FluorChem Q system (Biozym, Oldendorf, Germany) after being treated with Luminata™ Classico Western HRP substrate (Merck Millipore, Darmstadt, Germany) for 3 min. ImageJ was used to calculate protein expression [36]. Original, uncropped Western blot membranes can be found in Appendix A.

### 2.12. T-ALL Xenograft Zebrafish Model

Adult zebrafish of the wild-type AB strain were ordered from the China Zebrafish Resource Center, Institute of Hydrobiology, China Academy of Science (Wuhan, China) and officially approved by the Association for Assessment and Accreditation of Laboratory Animal Care International (SYXK 2012-0171). 48 h post-fertilization, natural pair-mating procreated zebrafish larvae were grown in an aquaculture space with a photoperiod of 14 h day/10 h night in a row. Zebrafish larvae were nourished with fry flakes (one portion) and live brine shrimps (two portions) every day.

To establish a T-ALL xenograft zebrafish model, CCRF-CEM cells were labelled with red fluorescence CM-Dil at a ratio of 1:1000. After 48 h post-fertilization, CCRF-CEM labelled cells (200 cells/zebrafish) were microinjected into the larvae yolk sac. Following that, 24 h post-tumor growth, the success of the established model was verified with a fluorescent microscope (AZ100, Nikon, Tokyo, Japan). The zebrafish bearing the labelled CCRF-CEM cells were then subjected to different adapalene concentrations, or the positive control cis-platinum for 24 h. The fluorescence intensity (Fi) of the tumor mass was measured in each single zebrafish and the inhibitory rate was determined using the following equation [37]:% inhibitory rate = [1 − (Fi _treated group_/Fi _negative control group_)] × 100%

## 3. Results

### 3.1. In Silico Drug Screening

In the field of drug development, virtual screening arose as a potent computational method to swipe vast libraries of small molecules for novel chemical hits with desirable features [38]. In the present study, the FDA-approved Zinc 15 database was screened to assess the in silico binding strength of its compounds toward c-MYC using the PyRx0.9 virtual screening tool. More than 100 compounds with predicted binding affinities of less than −6.10 kcal/mol were identified by PyRx (Appendix A). Subsequently, molecular docking (blind mode) was performed using AutoDock 4.2.6 to validate the outcomes of the virtual drug screening and to examine the preferential binding site of the ligands. The top 100 ligands with the lowest binding energies (LBEs) are depicted in Figure 1B (the top 117 ligands with the lowest binding energies (LBEs) are also shown in Appendix A). Moreover, a significant correlation (R^2^ = 0.9977) was observed between the LBE (kcal/mol) and the predicted pKi values (µM) (Figure 1C).

Among the top 20 compounds, we focused on atovaquone and adapalene as potential c-MYC inhibitor candidates. The LBE values of atovaquone and adapalene were −7.35 kcal/mol and −7.27 kcal/mol, respectively (Figure 1D). Interestingly, atovaquone and adapalene exerted stronger binding affinities to c-MYC, if compared to the known c-MYC inhibitors, 10058-F4 and 10074-G5 (LBEs of −4.92 kcal/mol and −6.24 kcal/mol, respectively) (Figure 1D). It is noteworthy that atovaquone bound to the same binding cavity as 10058-F4. Moreover, adapalene and 10074-G5 bound to the same binding pocket, and the amino acid residue PRO938 was involved in binding to both substances (Figure 1D and Figure 2).

All compounds had one amino acid involved in H-bonding except for adapalene, which had two amino acids engaged in H-bonding (Figure 1D). The molecular docking further showed that atovaquone and adapalene strongly bound to the bHLHZip domain of c-MYC.

### 3.2. Effectiveness of Atovaquone and Adapalene against HL60 and U266 Cells

Based on the COSMIC database (https://cancer.sanger.ac.uk/cosmic, accessed on 25 October 2019), HL60 acute myeloid leukemia cells (AML) overexpress c-MYC. On the contrary, Holien et al. (2012) showed that U266 MM cells lack c-MYC expression [39]. To assert these findings, c-MYC protein expression in both cell lines was assessed by us. Our results revealed that HL-60 truly overexpressed c-MYC, while U266 slightly expressed c-MYC. Thus, our U266 findings were contradictory to those of Holien et al. (2012). In our hands, the c-MYC expression is 3× greater in HL-60 than in U266 cells (Figure 3A). In an attempt to examine the specificity of atovaquone and adapalene to c-MYC, we evaluated the cytotoxic effect of atovaquone and adapalene in HL60 and U266 cells, respectively. As expected, HL60 cells overexpressing c-MYC were more susceptible to adapalene and atovaquone than U266 cells that had only a modest c-MYC expression. Moreover, HL60 displayed a greater sensitivity to adapalene (IC_50_ = 1.19 ± 0.17 µM) compared to atovaquone (IC_50_ = 19.54 ± 2.26 µM) (Figure 3B). Therefore, we proceeded with our experiments with adapalene as a potential c-MYC inhibitor candidate.

**Figure 2 cancers-15-04136-f002:**
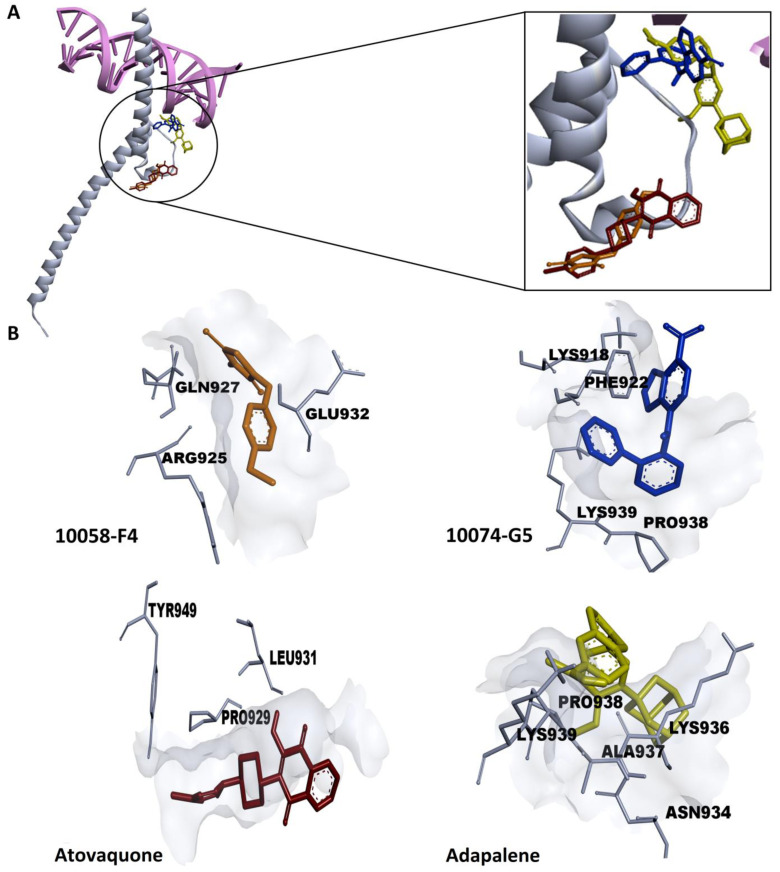
Molecular docking of atovaquone, adapalene, and the known c-MYC:MAX interaction inhibitors: 10058-F4 and 10074-G5. (**A**) Crystal structure of c-MYC (grey) bound to E-Box DNA (pink) (PDB ID: 1NKP). Atovaquone and adapalene shared the same binding cavity as the known inhibitors. (**B**) Amino acids residues expected to be involved in the interaction between c-MYC (grey) and 10058-F4 (orange), 10074-G5 (blue), atovaquone (red), as well as adapalene (yellow), were generated using Discovery Studio Visualizer software (version v.21.1.0.20298).

### 3.3. Adapalene Strongly Bound to c-MYC

MST is a sensitive approach to confirm the interaction between an unlabeled ligand and a labelled protein [40]. Therefore, we performed MST to verify the in silico predicted binding of adapalene to c-MYC. The fluorescence-labeled recombinant human c-MYC protein was titrated against several adapalene concentrations (Figure 3C). A fit curve was accomplished in accordance with the law of mass action, and adapalene was shown to robustly bind to c-MYC with a K_d_ value of 3.05 µM. The MST findings were in line with those of the in silico results.

### 3.4. Adapalene Reduced the Viability of Hematological Cancer Cells

c-MYC plays a crucial role in MM pathophysiology [41]. Given that adapalene might be a novel potential c-MYC inhibitor that has not yet been reported as a therapy for hematological malignancies, we examined the cytotoxic effect of adapalene in MM, T-ALL, and PBMCs using the resazurin cell viability assay. After 72 h of treatment, all evaluated MM cell lines showed substantial sensitivity towards adapalene. The most sensitive cell line, AMO1, had an IC_50_ value of 1.76 ± 0.39 µM, while the least sensitive cell line, JJN3, had an IC_50_ value of 9.10 ± 1.85 µM (Figure 4 and Table 1). Moreover, CCRF-CEM and CEM/ADR5000 cells were sensitive to adapalene to nearly a similar extent (IC_50_ = 1.83 ± 0.46 µM and 2.30 ± 0.09 µM, respectively). Therefore, multidrug-resistant CEM/ADR5000 cells were not cross-resistant to this drug (degree of resistance: 1.26). The concentration of adapalene needed to inhibit the viability of JJN3 cells (the least sensitive cell line) by 50% was lower than the concentration required to inhibit 50% of healthy leucocytes (IC_50_ = 36.72 ± 0.64 µM) (Figure 4 and Table 1), indicating at least some tumor specificity of this compound.

### 3.5. Adapalene Inhibited c-MYC Expression and Transcriptional Activity In Vitro

Based on the in silico and MST results, we wanted to further confirm the hypothesis that adapalene may be a c-MYC inhibitor. Therefore, we investigated whether adapalene impeded c-MYC expression and transcriptional activity. Adapalene treatment considerably and dose-dependently downregulated c-MYC expression (Figure 5A). In an effort to evaluate whether adapalene’s inhibitory effect was not just confined to c-MYC expression but might possibly also involve the inhibition of c-MYC transcriptional activity, a c-MYC reporter luciferase construct was transfected into HEK293 cells. These cells were then subjected to various concentrations of adapalene and 10058-F4, a known c-MYC inhibitor, respectively. Indeed, adapalene dose-dependently suppressed the transcriptional activity of c-MYC. Interestingly, low adapalene concentrations suppressed c-MYC activity more effectively than the established inhibitor 10058-F4, demonstrating the strong c-MYC inhibitory potential of adapalene (Figure 5B).

### 3.6. Cell Cycle Investigations in Adapalene-Treated AMO1 Cells

Many cytotoxic substances exert their anticancer activities by affecting the cell cycle progression of cancer cells [20]. To examine the impact on the cell cycle in AMO1 cells, we incubated the cells for 48 and 72 h with various adapalene concentrations (0.5 × IC_50_, 1 × IC_50_, and 2 × IC_50_). The percentage of AMO1 cells in the G2/M (indicating mitotic arrest) and sub-G0/G1 phases (indicating cell death) increased in direct proportion with increasing adapalene concentrations (Figure 6).

### 3.7. Adapalene Induced Mitotic Arrest as Detected by Phospho-Histone-3 (Ser10) Immunofluorescence

Phospho-histone-3 (Ser10) (p-H3(Ser10)) represents a cell cycle marker to assess cells in the late G2/M phase [42]. To confirm whether adapalene induced G2 or M arrest, we examined the effect of adapalene on the expression of p-H3(Ser10), a mitotic hallmark that is crucial in regulating the mitotic catastrophe [43]. Adapalene significantly increased the expression of p-H3(Ser10) in AMO1 cells after treatment for 48 h (Figure 7). Therefore, adapalene indeed induced the accumulation of AMO1 cells in the M-phase.

### 3.8. Adapalene Mitotic Arrest Involved the Distortion of the Tubulin Network as Detected by Immunofluorescence

Mitotic arrest is most likely caused by the suppression of microtubule polymerization because microtubules are essential for chromosomal separation during mitosis [44]. Considering that adapalene led to G2/M arrest, we examined its impact on the tubulin network. U2OS cells stably transfected with α-tubulin-GFP fusion construct were treated with adapalene (1 × IC_50_ and 2 × IC_50_) for 48 h, and the tubulin network was visualized by confocal microscopy. Control cells perfectly polymerized the tubulin network. This was demonstrated by the significant amount of tubulin that disseminated throughout the cytoplasm and produced a strong intracellular network. By contrast, the tubulin network appeared in a chaotic shape upon adapalene treatment. Adapalene expanded the tubulin mass surrounding the nucleus while decreasing the growth of microtubules at the borders. Additionally, adapalene-treated cells displayed fragile microtubules at their borders if compared to control cells (Figure 8). Comparing adapalene’s effect to positive the control drugs nocodazole (as polymerization inhibitor) and paclitaxel (as depolymerization inhibitor) [45] allowed us to conclude that adapalene prevented the polymerization of the microtubule network in a manner similar to nocodazole.

### 3.9. Adapalene Induced Apoptosis, Autophagy, and DNA Damage in AMO1 Cells

Our cell cycle results showed a dramatical dose-dependent increase in the fraction of cells in the sub-G0/G1 phase, indicating that adapalene promoted cell death. In an effort to determine the cell death mode(s) induced by adapalene, we studied the three main processes of cell death: apoptosis (type I), autophagy (type II), and necrosis (type III) [46]. First, we investigated apoptosis, a form of programmed cell death that most likely takes place in cancer cells treated with cytotoxic drugs. A FITC-conjugated annexin V/PI assay was used to distinguish between living, early apoptotic, late apoptotic, and necrotic cells. Annexin V is usually detected in early and late apoptosis. However, PI is detected in late apoptosis and necrosis. The positive control drug bortezomib effectively induced early and late apoptotic cells after 48 h treatment. Increasing the incubation period to 72 h considerably induced early apoptotic cells without significantly affecting the portion of late apoptotic cells. The number of early and late apoptotic cells significantly increased in a dose-dependent manner upon treatment with adapalene for 48 h. Interestingly, extending the incubation time to 72 h did not considerably change the proportion of apoptotic cells. Moreover, necrotic cells were not detected at all (Figure 9), indicating that adapalene induced apoptosis in a comparable manner after 48 and 72 h.

Next, we examined autophagy as a type II cell death modality. Autophagy detection using flow cytometry revealed that the untreated control cells displayed a weak fluorescence signal. However, the intensity of this signal increased proportionally with increasing adapalene concentrations, reaching an approximately 1.7-fold increase with 2 × IC_50_ after 48 h treatment. Interestingly, adapalene induced autophagy more significantly than rapamycin, a common autophagy inducer (Figure 10A). Adapalene continued to trigger autophagy after 72 h treatment, albeit to a lower extent if compared to 48 h post-treatment (Figure 10B), suggesting a potential switch to another type of cell death. To strengthen our conclusions, we further examined autophagy by detecting the autophagic markers Beclin1 and LC3BII by Western blotting after 48 h treatment. Western blot analyses showed that the protein expression level of Beclin1 and LC3BII increased if AMO1 cells were treated with 0.5 × IC_50_ and 1× IC_50_ of adapalene. However, Beclin1 and LC3BII expression diminished upon treatment with 2 × IC_50_ (Figure 10C), indicating that adapalene prompted autophagy.

It has been known for many years that DNA double-strand breaks and c-MYC dysregulation are tightly related [47]. Given that γH2AX is a marker of DNA double-strand breaks [48], we determined the γH2AX expression 48 h post-adapalene treatment. Western blot analysis revealed that the γH2AX expression effectively increased in a dose-dependent manner in AMO1 cells (Figure 10C), suggesting that adapalene triggered DNA damage.

### 3.10. Anticancer Activity of Adapalene In Vivo

The anti-tumor efficacy of adapalene has been confirmed in vivo using murine models [20,22]. In this study, a CCRF-CEM xenograft tumor model was established in larvae zebrafish, and the fluorescence intensity was measured to calculate the tumor inhibition rate after 24 h treatment. Comparing the positive control imatinib mesylate with the untreated group revealed that imatinib mesylate significantly reduced the intensity of the fluorescent CCRF-CEM tumors (Figure 11). The same was true for adapalene, which reduced the growth of CCRF-CEM tumors with an inhibitory rate of 38.23% at 3.47 nM. Even though adapalene has been administrated at an extremely low dose, it nonetheless demonstrated a higher inhibitory rate than the positive control drug. Our investigations revealed that adapalene possessed anti-tumor properties in a larvae zebrafish model.

## 4. Discussion

Despite substantial improvements in MM treatment, this disease is currently regarded as incurable due to relapse and resistance to chemotherapy, highlighting the need for new treatments with alternative approaches [13]. The proto-oncogene c-MYC is considered as an appealing therapeutic target for MM treatment since its dysregulation is a distinctive genetic feature of MM. Numerous strategies suggested so far focus on c-MYC in MM. However, the pharmacokinetics and efficacy of most of these techniques were unsatisfactory, rendering the direct inhibition of c-MYC very challenging in the context of ligand discovery [13]. This fact urged us to search for novel compounds that directly inhibit c-MYC. Therefore, we performed targeted screening of the FDA-approved drug library using in silico tools (virtual screening and molecular docking). Vitamin A, as well as its natural and synthetic analogues, known as retinoids, are essential for several vital processes such as pattern formation in embryogenesis, bone development, hematopoiesis, differentiation, metabolism, reproduction, and vision [49]. Retinoids have shown a strong antiproliferative effect and have been beneficial in the treatment of a number of human disorders, including cancer [50]. Due to their strong gene regulation capability and receptor-binding affinities, retinoids have been widely claimed to possess anticancer activities [51]. In fact, they have demonstrated promising therapeutic results in clinical studies for solid tumors as prostate cancer, basal cell skin cancer, head and neck cancer, renal cancer, cervical cancer, and breast cancer [52]. Moreover, all-trans-retinoic acid (ATRA)-based differentiation therapy has also been very effective in treating acute promyelocytic leukemia (APL) [53]. In comparison to natural retinoids, synthetic retinoids have demonstrated greater selectivity, stronger efficacy, and reduced toxicity [51]. Adapalene belongs the third generation of retinoids [16]. Although it was originally a topical treatment of acne vulgaris [54], we found that it strongly bound to the c-MYC bHLHZip domain at the same binding pocket as the known inhibitor 10074-G5. This binding pocket represents the MYC/MAX interaction site which is essential for the function of c-MYC. As adapalene shared the same binding pocket with 10074-G5, it is presumed that they exhibit similar modes of action. They inhibit c-Myc/Max heterodimer formation and thus inhibit its transcriptional activity. Interestingly, adapalene showed higher affinity to c-MYC if compared to the positive controls 10058-F4 and 10074-G5. Hence, adapalene might be a novel candidate c-MYC inhibitor, and it may be a third-generation retinoid repurposed for the treatment of MM. To validate this hypothesis, we examined the molecular interaction of adapalene and c-MYC using MST. Indeed, adapalene robustly bound to c-MYC (K_d_ = 3.05 µM), and these findings were in line with those of the in silico analysis. Next, we investigated the sensitivity of a panel of nine MM cell lines as well as drug-sensitive CCRF-CEM and multidrug-resistant CEM/ADR5000 leukemia cells towards adapalene. All MM cell lines responded to adapalene with different degrees of sensitivity (IC_50_ values from 1.76 ± 0.39 μM to 9.10 ± 1.85 μM). Since AMO1 was the most sensitive cell line (IC_50_ = 1.76 ± 0.39 μM), we investigated the mechanisms of action of adapalene in these cells. It is known that CEM/ADR5000 cells, overexpressing the multidrug-resistance-mediating ABC-transporter P-glycoprotein, extrude chemotherapeutic drugs out of cells resulting in chemotherapy failure [55,56]. According to our findings, CEM/ADR5000 cells and their sensitive counterparts both demonstrated a similar level of sensitivity to adapalene. Furthermore, the resistance ratio was just 1.26, which is far lower than that of doxorubicin, a common chemotherapeutic medication with a cross-resistance ratio much more than 1000 [30]. These findings suggested that adapalene is not involved in the cross-resistance profile conferred by P-glycoprotein. The concentration of adapalene needed to inhibit the proliferation of 50% of human peripheral blood mononuclear cells (PBMCs) was much higher than those of all tested leukemia and MM cell lines. This suggests that the concentration of adapalene required to kill leukemia and MM cells might be also reached in patients without harming healthy hematopoietic cells. To the best of our knowledge, we are the first to demonstrate that the retinoid adapalene inhibited MM cells by suppressing c-MYC, suggesting a novel mode of action of adapalene not as an anti-inflammatory but as a potential anticancer agent.

The oncogene c-MYC is overexpressed in several cancer types, and it is a causative factor of at least 40% of malignancies. On the other hand, c-MYC expression is firmly controlled in healthy cells [57]. c-MYC primarily acts as a transcriptional regulator, modulating the expression of genes that are involved in a variety of cellular activities, including transcription, translation, cell division, metabolism, cell differentiation, DNA repair, apoptosis, autophagy, immune response, and stem cell growth [58]. Its contribution to oncogenesis was initially discovered due to its homology to an avian retrovirus and was highly expressed in Burkitt’s lymphoma because of the t(8;14) translocation [59,60]. Moreover, c-MYC deregulation is one of the main features in MM progression. Although targeting c-MYC is challenging, numerous investigations demonstrated the potential significance of c-MYC inhibition for cancer therapy [61]. Currently, no MYC inhibitor has been clinically established, but significant efforts are globally ongoing to realize this possibility for cancer patients [62]. In this context, verifying that adapalene inhibits c-MYC may be a glimmer of hope, especially because it is an FDA-approved drug, and its toxicity profiles, pharmacokinetics, and pharmacodynamics are well established. To further validate adapalene as c-MYC inhibitor, we performed a c-MYC reporter assay. Adapalene significantly reduced the DNA-binding activity of c-MYC. It is interesting to note that this inhibitory impact was substantially stronger than the prominent c-MYC inhibitor 10058-F4. Additionally, adapalene concentrations required to inhibit c-MYC were far lower than that of 10058-F4. Furthermore, adapalene significantly and dose-dependently reduced the expression of c-MYC. Taken together, our in vitro results were in line with those of the in silico data and conclude that adapalene may indeed inhibit c-MYC expression and its transcriptional activity.

c-MYC activates the expression of genes that are essential regulators of the cell cycle, including E2F transcription factors, cyclins, and cyclin-dependent kinases (CDKs). Additionally, c-MYC inhibits Cdk inhibitors such as p21 and p27 by activating SKP2, which is involved in p27 degradation, or by inhibiting CDKN1A, which codes for p21. Furthermore, c-MYC activates CUL1, a component of the E3 ubiquitin ligase complex necessary for the effective ubiquitination and destruction of p27 [63]. Moreover, inhibiting c-MYC caused human myeloid and lymphoid cells arrest the cell cycle [64]. Given the important role of c-MYC for cell cycle control, we examined the cell cycle progression of adapalene-treated AMO1 cells after 48 and 72 h treatment. Adapalene arrested AMO1 cells in the G2/M phase besides an increase in the sub-G0/G1 phase after 48 and 72 h treatment. To investigate the molecular mechanisms underlying the G2/M phase arrest, immunofluorescence microscopy was performed to determine the number of adapalene-treated AMO1 cells that were positive for p-H3(Ser10). Indeed, adapalene treatment caused AMO1 cells to preferentially assemble in the M-phase. Given that Ser10 phosphorylation occurs in early prophase, is maintained during metaphase, is decreased during anaphase, and vanishes entirely during telophase before or at the start of chromosomal decondensation [65], an increase in the number of positive p-H3(Ser10) represents an indication of adapalene’s ability to hamper mitotic exit in AMO1 cells. Usually, mitotic catastrophe is the most prevalent outcome of extended mitotic arrest. It is a mechanism that senses mitotic failure and responds by initiating irreparable, anti-proliferative cell death [66]. This may account for the time- and dose-dependent increase of cells in the sub-G0/G1 phase. Our results unexpectedly differed from those of other studies where adapalene induced S-phase arrest in a prostate cancer cell, a triple-negative breast cancer cell, and melanoma cells [22,23], or G1-phase arrest in colorectal cancer cells [19]. It seems that phase-specific cell cycle arrest is tumor cell line-specific.

The accumulation of cells in the G2/M phase may indicate that G2/M mediators are affected by adapalene. Therefore, we studied the impact of adapalene on the microtubule network. Moreover, previous studies demonstrated that c-MYC interacts with α-tubulin and with polymerized microtubules in vitro and in vivo [67]. Microtubules have also been hypothesized to serve as a cytoplasmic repository for c-MYC proteins [68]. Additionally, several drugs that disrupt microtubule dynamics decreased c-MYC expression by reducing the cytoplasmic c-MYC reservoir [69]. Thus, it was fascinating for us to prove whether this reciprocal aspect was true. Microtubules are highly active cytoskeletal structures that are important for many cellular functions such as vesicle transport, cell division, and intracellular organization. Microtubule-targeting agents inhibit the dynamics of microtubules, resulting in a slowdown or blockage of mitosis. This block, which takes place in the G2/M phase, may trigger cell death. Confocal microscopy of adapalene-treated U2OS cells stably expressing α-tubulin-GFP showed an aberrant microtubule arrangement. In fact, adapalene expanded the tubulin mass, surrounding the nucleus while decreasing the growth of microtubules at the borders. Additionally, adapalene-treated cells displayed fragile microtubules at their borders if compared to control cells. To the best of our knowledge, we are the first to decipher that adapalene effectively disrupted the microtubule network, which may account for its cytotoxicity.

Several authors investigated the essential role of c-MYC in DNA damage signaling [70]. Responses to intracellular stress signals as DNA damage resulted in decreased c-MYC levels due to proteasomal degradation [71]. Moreover, DNA damaging agents (e.g., topoisomerase II inhibitors) or ionizing radiation induced DNA damage as well as c-MYC inhibition in MCF-7 breast cancer cells [72,73]. Furthermore, the knockdown of c-MYC in HeLa-630 cells limited their capacity to repair double-strand breaks following ionizing irradiation [74]. Based on the crucial role of c-MYC in DNA damage repair, studies have demonstrated a relationship between c-MYC and γH2AX. Phosphorylated c-MYC displayed a strong co-localization with γH2AX and the phosphorylated DNA-dependent protein kinase, catalytic subunit at the corresponding S2056 cluster. Consequently, the inhibition of c-MYC resulted in double-strand breaks and an increase in γH2AX [74]. Taking into consideration this relationship between c-MYC and γH2AX, we investigated the effect of adapalene on the protein expression level of γH2AX. Intriguingly, adapalene treatment resulted in a dose-dependent increase in γH2AX, suggesting that the ability of adapalene to inhibit MM growth may be due to its capability to induce DNA damage. Our results were in line with other studies, showing that adapalene induced apoptosis through DNA damage in melanoma cells [20]. It also reduced the growth of prostate cancer cells by triggering DNA damage [22].

An eminent function of c-MYC is its ability to sensitize cells to apoptosis. Dysregulated c-MYC expression accompanied by anti-proliferative signals initiates apoptosis [75]. Therefore, we examined whether adapalene-mediated cytotoxicity in AMO1 cells was associated with apoptosis. Our results revealed that adapalene induced both early and late apoptosis. Our findings concurred with those of another study showing that the suppression of c-MYC by CPI-0610, a small-molecule bromodomain and extra-terminal protein inhibitor, induced G0/G1 arrest as well as apoptosis in MM, leading to cancer regression [76]. In addition, adapalene promoted apoptosis in triple-negative breast cancer cells, melanoma cells, colorectal cancer cells, bladder cancer cells, hepatoma cells, prostate cancer cell, and ovarian cancer cell [17,20,21,22,23,77,78]. To the best of our knowledge, we are the first to demonstrate that adapalene-mediated c-MYC inhibition induced apoptosis in MM.

Autophagy is a vital catabolic cellular process in which deteriorated organelles, misfolded proteins, and cellular components are digested and recycled by lysosomes. Autophagy is essential for maintaining cellular homeostasis and allowing cells to adjust to harsh conditions. Additionally, it controls both cell survival and death [79]. The microtubule-associated protein light chain 3 (LC3) and Beclin 1 play essential roles in autophagy. Beclin 1 is a member of the phosphatidylinositol 3-kinase complex (PI3K) type III and is necessary for autophagic vesicle formation. LC3 is activated by a ubiquitination-like reaction mediated by Atg3 and Atg7. Initially, LC3 is cleaved into cytosolic LC3-I which is further changed into a membrane-bound form LC3-II. LC3-II is then attracted to autophagosomes. Therefore, it is considered a special autophagy marker [80]. To decipher whether adapalene triggers autophagy in AMO1 cells, we studied the protein expression level of LC3 and Beclin 1. The fact that LC3-II and Beclin 1 were both upregulated upon adapalene treatment approved that the latter induced autophagy in AMO1 cells. Those findings were further confirmed by an autophagy assay. As far as we know, no previous studies reported the ability of adapalene to trigger autophagy. Several groups demonstrated the interplay between c-MYC and the regulation of autophagy. For instance, 4-O-methyl-ascochlorin inhibited c-MYC and triggered autophagy in leukemic cells [81]. The inhibition of c-MYC by 10058-F4 led to the downregulation of miR-150 and restored the miR-150-mediated autophagy defect in NSCLC [82]. Our findings might also indicate that the induction of autophagy could be accredited to c-MYC inhibition. Surprisingly, the expression of Beclin 1 and LC3-II dropped off at high concentrations (2 × IC_50_). This could be explained by a shift to another cell death as apoptosis. These observations were not detected in the autophagy assay as western blot is more sensitive and detects even minimal changes in protein expression. Numerous reports proved that Beclin 1 is a caspase 3 substrate. Upon caspase 3 activation, Beclin 1 is cleaved and becomes unable to induce autophagy. However, the C-terminal fragment (Beclin 1-C) may relocate to the mitochondria and make cells more susceptible to apoptosis [83]. A reduction of Beclin 1 prompted CED-3/caspase-dependent programmed cell death in *C. elegans* [84]. Moreover, it has been suggested that LC3 can act as a promoter of apoptosis [83].

Several authors verified adapalene’s in vivo anti-tumor activity in mice [19,22]. Recently, the zebrafish model proved to be an effective in vivo model for both drug development and toxicity evaluation [85]. In the current investigation, we demonstrated that adapalene inhibited tumor growth in vivo by utilizing a CCRF-CEM xenograft tumor zebrafish model.

One of the study’s shortcomings is the inability to examine the impact of adapalene on PBMCs derived from patients with hematological malignancies. In addition, a crucial tool for clinical translation is the analysis of adapalene in murine models of human MM. Therefore, we strive to select the appropriate MM murine model for specific pre-clinical research.

## 5. Conclusions

In conclusion, the knowledge gained from this work points to the potential of the third-generation retinoid adapalene to significantly suppress tumor growth via autophagic and apoptotic cell death. Furthermore, adapalene possesses a potent cytotoxic activity through c-MYC inhibition, tubulin network suppression, and DNA damage. The discovery of adapalene as a c-MYC inhibitor may significantly lower the drug development costs of new anticancer medications, providing further insights in the future of adapalene-based designs that could result in more effective and targeted innovative therapies for MM.

## Figures and Tables

**Figure 1 cancers-15-04136-f001:**
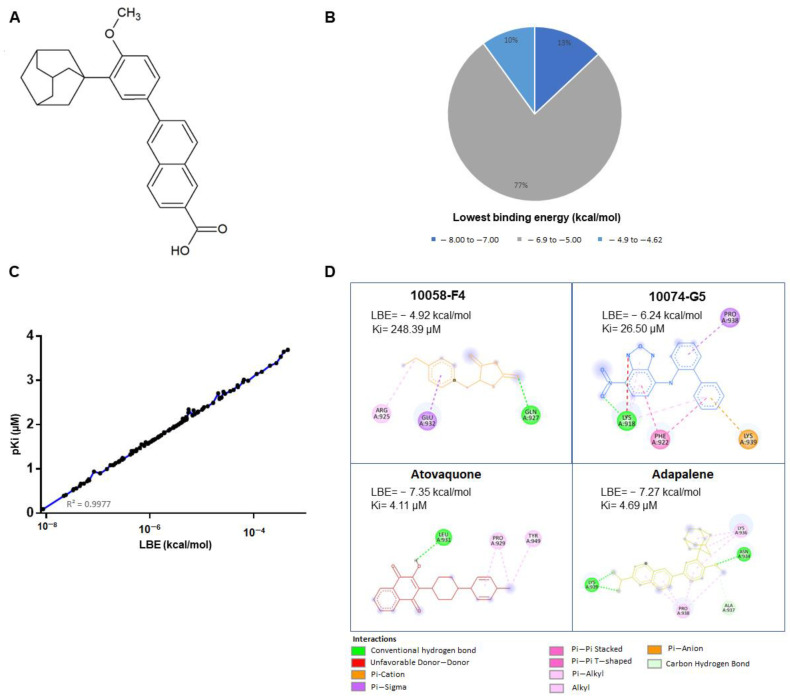
Molecular docking of candidate c-MYC inhibitors. (**A**) Chemical structure of adapalene. (**B**) Pie chart showing the percentage of FDA-approved ligands belonging to a particular lowest binding energy (kcal/mol) range. (**C**) Correlation coefficient representation of predicted inhibition constant (pKi) vs. lowest binding energies (LBE). (**D**) 2D representations of the various interactions between the amino acid residues of c-MYC and the corresponding compounds (10058-F4, 10074-G5, atovaquone, and adapalene) were generated using Discovery Studio Visualizer software (version v.21.1.0.20298). The lowest binding energies (LBE) and the predicted inhibition constant (Ki) of each compound were acquired from AutoDockTools 4.2.6.

**Figure 3 cancers-15-04136-f003:**
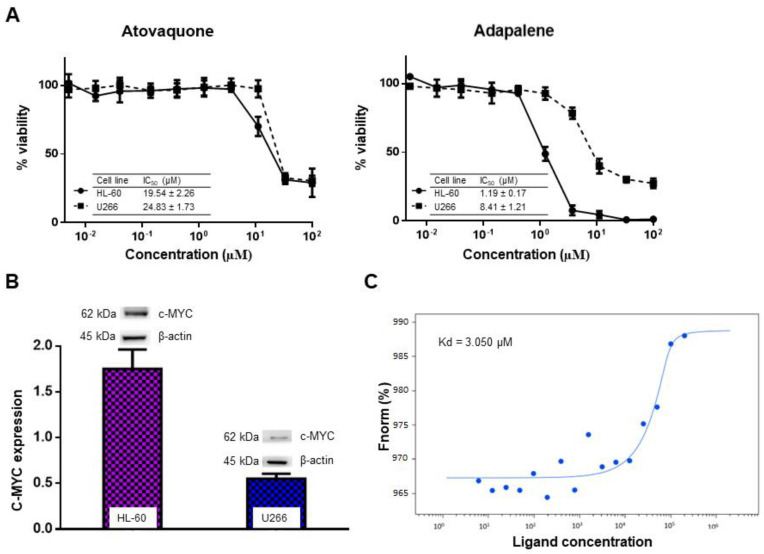
Specificity of atovaquone and adapalene towards c-MYC. (**A**) Assessment of c-MYC protein expression in HL60 and U266 cells by western blot. c-MYC protein expression was three times higher in HL-60 than in U266 cells. The bar diagram was obtained by calculating the mean value ± SD of three independent experiments. (**B**) The cytotoxic effect of atovaquone and adapalene in HL-60 and U266 cells. HL60 cells were more susceptible to adapalene and atovaquone than U266 cells. HL60 cells displayed a greater sensitivity to adapalene than to atovaquone. Adapalene showed higher specificity to c-MYC. (**C**) Binding of adapalene to recombinant human c-MYC as measured by microscale thermophoresis (MST). The fluorescence signal changed depending on the concentration as shown by the binding curve. The law of mass action was used to obtain fitness. MST was conducted with 70% LED power and 10% MST power.

**Figure 4 cancers-15-04136-f004:**
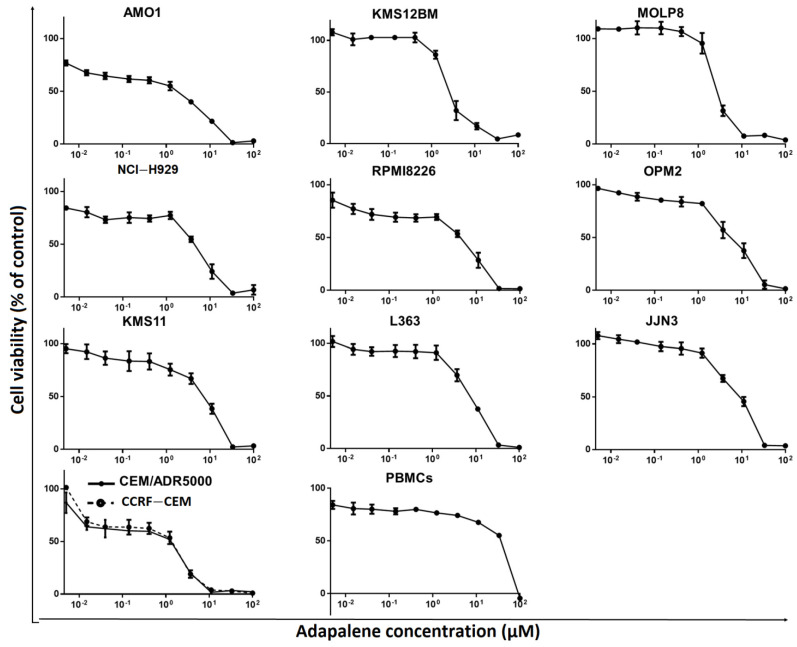
Adapalene’s cytotoxic effect in various human MM cell lines (AMO1, KMS12BM, MolP8, NCI-H929, RPMI8226, OPM2, KMS11, L363, and JJN3), in human T-ALL cells (CEM/ADR5000 and CCRF/CEM), and in peripheral blood mononuclear cells (PBMCs). Each value indicates the mean value ± SD of three distinct experiments, each of which had six replicates.

**Figure 5 cancers-15-04136-f005:**
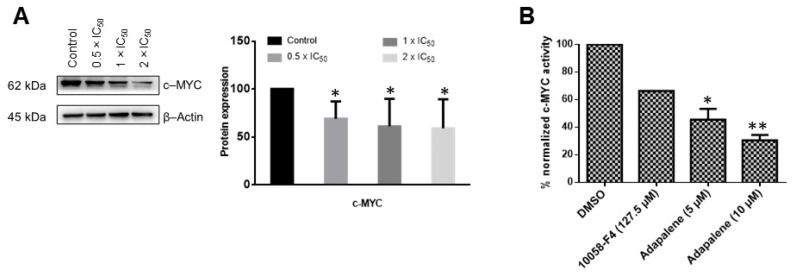
Adapalene affected c-MYC expression and transcriptional activity. (**A**) Effect of adapalene (0.5 × IC_50_, 1 × IC_50_, and 2 × IC_50_) on c-MYC protein expression in AMO1 cells as displayed by western blot. The bar diagram was created by calculating the mean value ± SD of three independent experiments. The uncropped bolts are shown in Appendix A. (**B**) Effect of adapalene and 10058-F4, a known c-MYC inhibitor, on c-MYC transcriptional activity. The bar diagram represents the percentage of normalized c-MYC activity in HEK293 cells transiently expressing a c-MYC-luciferase reporter construct. It was created by calculating the mean value ± SD of two independent experiments. * *p* < 0.05, ** *p* < 0.01, if compared to negative control.

**Figure 6 cancers-15-04136-f006:**
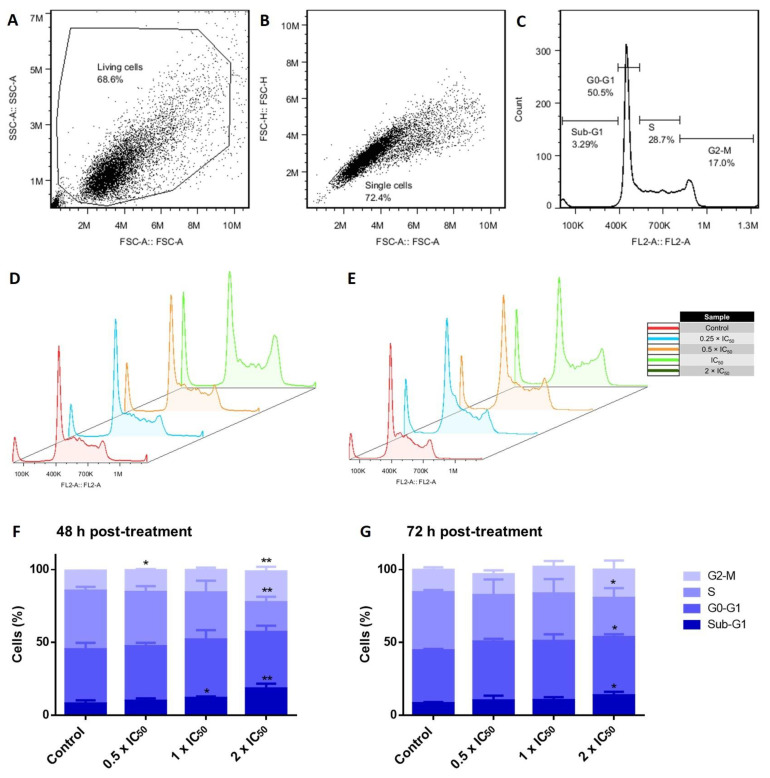
Cell cycle perturbations of AMO1 cells by adapalene. (**A**–**C**) Gating scheme: (**A**) The first gate (SSC-A vs. FSC-A) intends to get rid of the debris. (**B**) The second gate (FSC-H vs. FSC-A) aims to select only single cells. (**C**) Count vs. FL2-A represents cell cycle histograms. (**D**,**E**) DNA histograms of AMO1 exposed to adapalene (0.5 × IC_50_, 1 × IC_50_, and 2 × IC_50_) for 48 h (**D**) and 72 h (**E**). The histograms were obtained by flow cytometry using an excitation and an emission wavelength of 488 and 530 nm, respectively. (**F**,**G**) The bar diagrams showing the distribution of AMO1 cells treated with adapalene in the distinct phases of the cell cycle after 48 h (**F**) and 72 h (**G**) were created by calculating the mean value ± SD of two independent experiments. * *p* < 0.05, ** *p* < 0.01 if compared to negative control.

**Figure 7 cancers-15-04136-f007:**
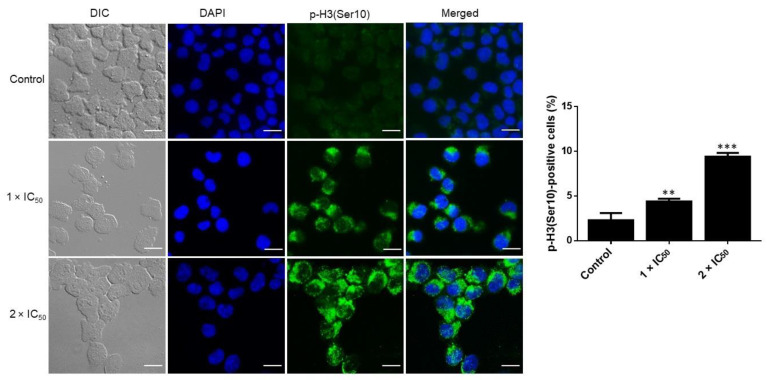
Accumulation of positive p-H3(Ser10) in AMO1 cells treated with adapalene (1 × IC_50_ and 2 × IC_50_) or DMSO (control) for 48 h. Anti-phospho-histone H3 (Ser10) clone 3H10, FITC-conjugated antibody (green), was used for immunostaining AMO1 cells. DAPI (blue) was used for nuclear staining. An AF7000 widefield fluorescence microscope at 40× magnification (scale bars = 20 µm) was used for imaging. The bar diagram represents the percentage of positive p-H3(Ser10). ** *p* < 0.01, *** *p* < 0.001 if compared to negative control.

**Figure 8 cancers-15-04136-f008:**
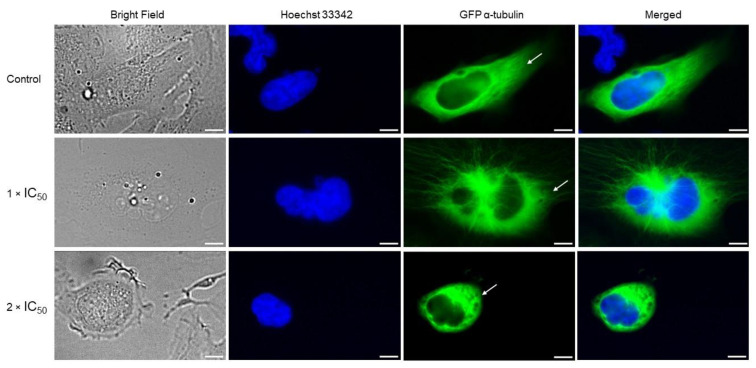
Chaotic microtubule network in U2OS cells stably transfected with GFP-α-tubulin protein upon adapalene treatment (1 × IC_50_ and 2 × IC50) for 48 h. U2OS cells were fixed with 4% paraformaldehyde. Micrographs were snapped 48 h after treatment with DMSO or adapalene treatment (1 × IC_50_ and 2 × IC_50_). DAPI (blue) was used for nuclear staining. The peripheral tubulin masses are pointed out with white arrows. An AF7000 widefield fluorescence microscope at 40× magnification (scale bars = 7 µm) was used for imaging.

**Figure 9 cancers-15-04136-f009:**
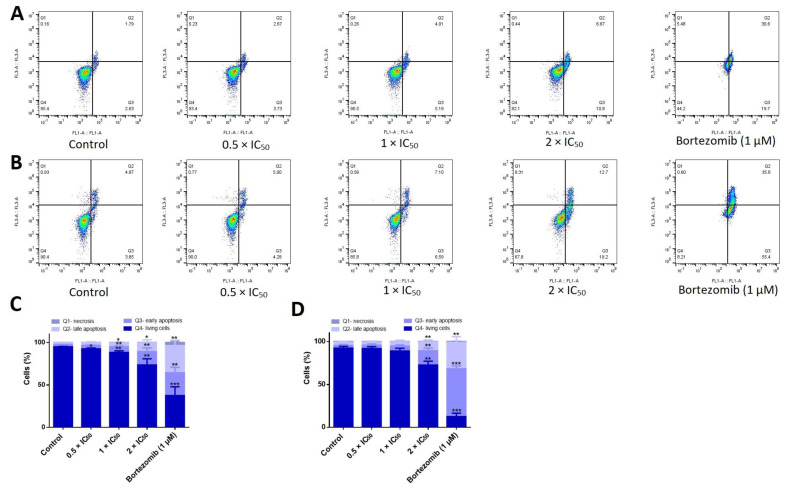
Induction of apoptosis in adapalene-treated AMO1 cells. (**A**,**B**) Apoptosis was examined in AMO1 cells treated with adapalene after 48 h (**A**) and 72 h (**B**) using annexin V/PI staining. Q4 represents viable cells (−) annexin V/(−) PI; Q3 represents early apoptotic cells (+) annexin V/(−) PI; Q2 represents late apoptotic cells exhibit annexin V (+)/PI (+); Q1 represents necrotic cells (−) annexin V/(+) PI. Treatment with adapalene at increasing concentrations greatly enhanced the percentage of early and late apoptotic cells after 48 and 72 h. (**C**,**D**) The bar diagrams showing the distribution of AMO1 cells in the different quadrants after 48 h (**C**) and 72 h (**D**) were created by calculating the mean value ± SD of three trials conducted at distinct times. * *p* < 0.05, ** *p* < 0.01, *** *p* < 0.001 if compared to negative control.

**Figure 10 cancers-15-04136-f010:**
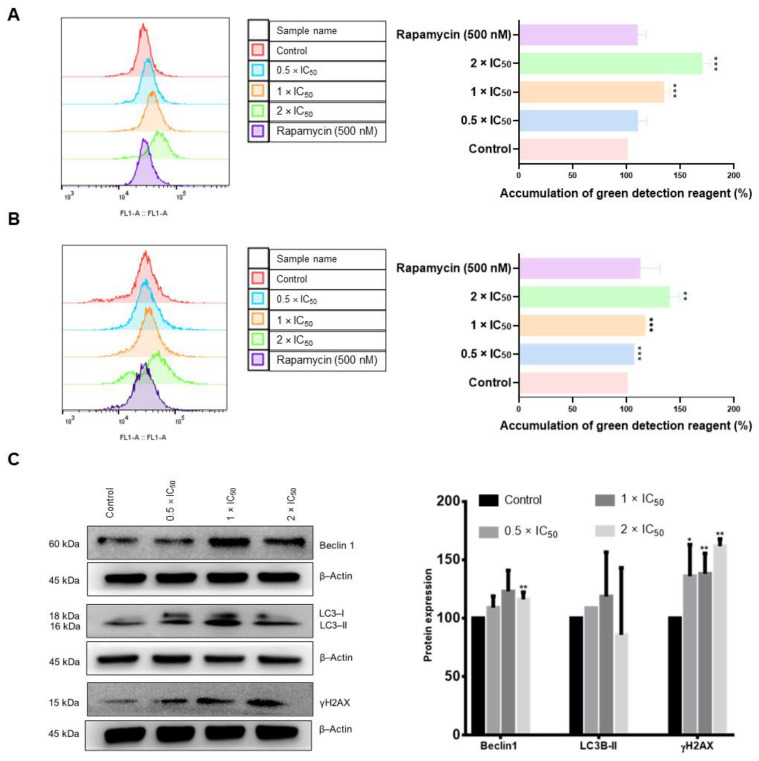
Induction of DNA damage and autophagy in adapalene-treated AMO1 cells. (**A**,**B**) Autophagy was examined in AMO1 cells treated with adapalene after 48 h (**A**) and 72 h (**B**) using green detection reagent. The histograms overlays were obtained by flow cytometry using an excitation and an emission wavelength of 463 and 534 nm, respectively. The bar diagrams were created by calculating the mean value ± SD of three trials conducted at distinct times. (**C**) Effect of adapalene (0.5 × IC_50_, 1 × IC_50_, and 2 × IC_50_) on the protein expression levels of γH2AX, Beclin 1, and LC3-II in AMO1 cells as detected by western blot. The bar diagram was created by calculating the mean value ± SD of three trials conducted at distinct times. * *p* < 0.05, ** *p* < 0.01, *** *p* < 0.001 if compared to negative control. The uncropped bolts are shown in Appendix A.

**Figure 11 cancers-15-04136-f011:**
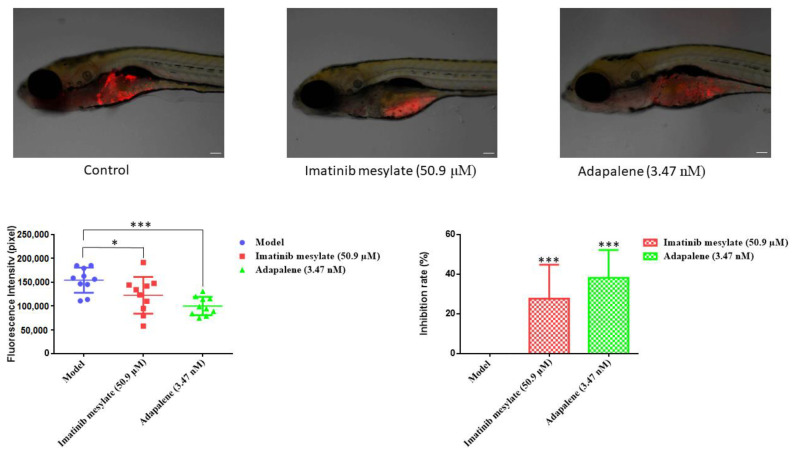
Examination of the acute cytotoxicity (fluorescence intensity and inhibition rate) of CCRF-CEM xenograft tumor zebrafish model (*n* = 10 zebrafish) treated with adapalene. The tumor mass of the CCRF-CEM was highlighted by the red fluorescence. Images were captured at a 60× magnification (scale bars = 100 µm). If compared to the model, * *p* < 0.05, *** *p* < 0.001 are significant.

**Table 1 cancers-15-04136-t001:** Adapalene’s IC_50_ values in MM, drug-sensitive T-ALL, and their multidrug-resistant P-glycoprotein-overexpressing leukemia sublines, as well as healthy leukocytes.

Cell Type	Cell Line	IC_50_ (µM)
MM	AMO1	1.76 ± 0.39
	KMS12BM	2.61 ± 0.36
	MOLP8	2.69 ± 0.29
	NCI-H929	4.95 ± 1.21
	RPMI8226	4.97 ± 1.33
	OPM2	5.82 ± 2.07
	KMS11	7.22 ± 1.20
	L363	7.25 ± 0.73
	JJN3	9.10 ± 1.85
Leukemia	CCRF-CEM	1.83 ± 0.46
	CEM/ADR5000	2.30 ± 0.09
Normal leukocytes	PBMCs	36.72 ± 0.64

## Data Availability

The data presented in this study are available on request from the corresponding author. The data are not publicly available due to privacy.

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
