# Peer review of "In Silico, In Vitro, and In Vivo Investigations on Adapalene as Repurposed Third Generation Retinoid against Multiple Myeloma and Leukemia"

_cancers, 2023, doi:10.3390/cancers15164136_

Round 1
Reviewer 1 Report
Tracking #: cancers-2492161 – Boulos et al.
The manuscript entitled “In silico, in vitro, and in vivo investigations on adapalene as repurposed third generation retinoid against multiple myeloma and leukemia” attempts to address interesting new issues regarding the anti-cancer potential (myeloma and leukemia) of adapalene, a retinoid already used to treat acne. Finding new ways to treat these cancers is of prime importance.
The scientific and experimental approach to identifying adapalene as an inhibitor of the c-MYC protein and demonstrating the consequences of this inhibition and the relevance of using this compound in this context of both myeloma and leukemia is remarkable.
The title clearly announces what the reader will find in the paper and the introduction allows for a good understanding of the subject. The literature is extensive and often recent.
Authors present data from well done experiments that are very varied and complementary, and the experimental conditions are well defined and calibrated.
Overall, this study is very interesting and informative, and presents new and exciting results.
The following is a list of few points that should be clarified and/or improved before publication to make it sounder:
1/ Authors should indicate the other names also used for adapalene, i.e. CD271 and differin.
2/ In order to respect the order in which the figures are cited in the text and the construction of the manuscript, Figure 1 should be presented after Figures 2 and 3, with the exception of the structure (Fig1C) of adapalene.
3/ A decrease in the quantity of c-MYC in the presence of adapalene is reported. Do the authors have any idea of the underlying mechanism? Degradation of the protein? More generally, the authors should discuss the mechanism(s) by which adapalene inhibits c-MYC activity.
4/ The authors used a reporter system to measure the transcriptional activity of c-MYC and the effect of adapalene on this activity (decrease). How do the authors explain this adapalene-induced decrease? By the decrease in the level of c-MYC? Have CHiP assays been performed to measure the level of c-MYC on DNA?
5/ Error: On page 14, lines 485-491, the text should be removed because it corresponds to the legend for figure 7.
6/ Typing error p20 line 719: c-MAC instead of c-MYC.
7/ For greater convenience, authors should harmonise the font size in figure legends (ordinates and abscissae).
Author Response
Thank you for your constructive comments, which have helped to improve the quality of our manuscript. We addressed all of the comments in the recently submitted revised version.
1/ Authors should indicate the other names also used for adapalene, i.e. CD271 and differin.
The others name of adapalene were added in the text.
2/ In order to respect the order in which the figures are cited in the text and the construction of the manuscript, Figure 1 should be presented after Figures 2 and 3, with the exception of the structure (Fig1C) of adapalene.
Figure 1 in the previous submitted version is now Figure 3 in the recently submitted revised version.
3/ A decrease in the quantity of c-MYC in the presence of adapalene is reported. Do the authors have any idea of the underlying mechanism? Degradation of the protein? More generally, the authors should discuss the mechanism(s) by which adapalene inhibits c-MYC activity.
Adapalene strongly bound to the c-MYC bHLHZip domain at the same binding pocket as the known inhibitor 10074-G5. This binding pocket represents the MYC/MAX interaction site which is essential for the function of c-MYC. As adapalene shared the same binding pocket with 10074-G5, it is presumed that they exhibit similar modes of action. They inhibit c-Myc/Max heterodimer formation and inhibit its transcriptional activity. The inhibition of c-MYC transcriptional activity by adapalene was further validated with the MYC reporter assay. According to the literature, the small-molecule c-MYC inhibitor 10074-G5 also degraded c-MYC protein [1,2]. Our western blot results also revealed that c-MYC expression is downregulated, indicating that c-MYC is degraded upon adapalene treatment. The discovery of drugs that cause the degradation of their target proteins have been largely reported [3]. In our study, we showed that similarly to the known inhibitor 10074-G5, adapalene bound to c-MYC, inhibited its transcriptional activity, and degraded c-MYC protein.
4/ The authors used a reporter system to measure the transcriptional activity of c-MYC and the effect of adapalene on this activity (decrease). How do the authors explain this adapalene-induced decrease? By the decrease in the level of c-MYC? Have CHiP assays been performed to measure the level of c-MYC on DNA?
The MYC reporter assay is made to keep track of the MYC signaling pathway's activity in cells. The main goal of the MYC reporter assay is to investigate c-MYC transcriptional activity. A transfection-ready expression vector for c-MYC and a reporter vector for MYC luciferase are included in the kit. Inside the cells, c-MYC will bind to Max, translocate to the nucleus, and induce expression of the MYC luciferase reporter vector. This reporter contains the firefly luciferase gene under the control of multimerized MYC responsive elements located upstream of a minimal promoter. The MYC reporter is premixed with constitutively-expressing Renilla luciferase vector, which serves as an internal positive control for transfection efficiency. A reduction in luminescence indicates that c-MYC loss its transcriptional function. The results of the MYC reporter assay were in line with the in silico and in vitro results that revealed a robust binding between adapalene and c-MYC with a lowest binding energy (LBE) of −7.27 kcal/mol, a predicted inhibition constant (pKi) of 4.69 µM, and a dissociation constant (Kd value) of 3.05 µM. Unfortunately, we did not perform CHiP assays to measure the level of c-MYC on DNA as this technique is not established in our Lab.
5/ Error: On page 14, lines 485-491, the text should be removed because it corresponds to the legend for figure 7.
The legend of figure 7 was removed from the text.
6/ Typing error p20 line 719: c-MAC instead of c-MYC.
c-MAC is substituted by c-MYC
7/ For greater convenience, authors should harmonise the font size in figure legends (ordinates and abscissae).
The font size of ordinates and abscissae is harmonized in Figure 3, Figure 4, Figure 5, and Figure 11.
References
- Clausen, D.M.; Guo, J.; Parise, R.A.; Beumer, J.H.; Egorin, M.J.; Lazo, J.S.; Prochownik, E.V.; Eiseman, J.L., In vitro cytotoxicity and in vivo efficacy, pharmacokinetics, and metabolism of 10074-g5, a novel small-molecule inhibitor of c-myc/max dimerization. Journal of Pharmacology and Experimental Therapeutics 2010, 335, 715-727.
- Zhao, Q.; Assimopoulou, A.N.; Klauck, S.M.; Damianakos, H.; Chinou, I.; Kretschmer, N.; Rios, J.-L.; Papageorgiou, V.P.; Bauer, R.; Efferth, T., Inhibition of c-myc with involvement of erk/jnk/mapk and akt pathways as a novel mechanism for shikonin and its derivatives in killing leukemia cells. Oncotarget 2015, 6, 38934.
- Long, M.J.C.; Gollapalli, D.R.; Hedstrom, L., Inhibitor mediated protein degradation. Chemistry & biology 2012, 19, 629-637.

Reviewer 2 Report
The manuscript is well written and carefully presented, the discovery of adapalene as a c-MYC inhibitor may significantly lower the drug develop ment costs of new anticancer medications, it also provides further insights in the future of adapalene-based designs that could result in more effective and targeted innovative therapies for multiple myeloma.
The manuscript can be accepted in present form, except the AutoDock software was used to validate the outcomes of the virtual drug screening and to examine the preferential binding site of the ligands, and the method was not presented in detail and can be improved.
Author Response
Thank you for your comment. The method was improved by adding the parameters of the grid box.
Reviewer 3 Report
In silico, in vitro, and in vivo investigations on adapalene as repurposed third generation retinoid against multiple myeloma and leukemia by Boulos is an interesting manuscript, however it needs additional information.
Minor Comment
- The originals blots need label to identify in the main figures
Positive comments
- The authors screened 1,578 FDA-approved drugs
- c-MYC inhibitors are interesting
- adapalene arrested triple-negative breast cancer cells in the S-phase, inhibiting their proliferation
Author Response
Thank you for your positive comments. We sincerely value your positive reception of our manuscript.
1. Labelled uncropped original western blot membranes are submitted as supplementary figure (Figure S1)